# Fuzzy Evaluation Output of Taste Information for Liquor Using Electronic Tongue Based on Cloud Model

**DOI:** 10.3390/s20030686

**Published:** 2020-01-27

**Authors:** Jingjing Liu, Mingxu Zuo, Sze Shin Low, Ning Xu, Zhiqing Chen, Chuang Lv, Ying Cui, Yan Shi, Hong Men

**Affiliations:** 1College of Automation Engineering, Northeast Electric Power University, Jilin 132012, China; 2201700366@neepu.edu.cn (M.Z.); 2201600442@neepu.edu.cn (N.X.); 2201700405@neepu.edu.cn (Z.C.); 2201800451@neepu.edu.cn (C.L.); 2201600406@neepu.edu.cn (Y.C.); 2201500430@neepu.edu.cn (Y.S.); 2Department of Computer Science and Bioimaging Research Center, University of Georgia, Athens, GA 30602, USA; 3Biosensor National Special Laboratory, Key Laboratory for Biomedical Engineering of Education Ministry, Department of Biomedical Engineering, Zhejiang University, Hangzhou 310027, China; szeshin.low@zju.edu.cn

**Keywords:** electronic tongue, two-dimensional cloud model, fuzzy words, Chinese liquor

## Abstract

As a taste bionic system, electronic tongues can be used to derive taste information for different types of food. On this basis, we have carried forward the work by making it, in addition to the ability of accurately distinguish samples, be more expressive by speaking evaluative language like human beings. Thus, this paper demonstrates the correlation between the qualitative digital output of the taste bionic system and the fuzzy evaluation language that conform to the human perception mode. First, through principal component analysis (PCA), backward cloud generator and forward cloud generator, two-dimensional cloud droplet groups of different flavor information were established by using liquor taste data collected by electronic tongue. Second, the frequency and order of the evaluation words for different flavor of liquor were obtained by counting and analyzing the data appeared in the artificial sensory evaluation experiment. According to the frequency and order of words, the cloud droplet range corresponding to each word was calculated in the cloud drop group. Finally, the fuzzy evaluations that originated from the eight groups of liquor data with different flavor were compared with the artificial sense, and the results indicated that the model developed in this work is capable of outputting fuzzy evaluation that is consistent with human perception rather than digital output. To sum up, this method enabled the electronic tongue system to generate an output, which conforms to human’s descriptive language, making food detection technology a step closer to human perception.

## 1. Introduction

In the process of food detection, taste information is one of the important reference factors to reflect the characteristics of food. The Toko research group of Kyushu University in Japan developed a taste sensor based on PVC film, where the intensity of taste was reflected by the electrical signal. This work has provided a solid foundation for human to evaluate food taste objectively [1,2]. Since then, research on electronic tongue technology was widely explored, and it has become more advanced to be used in food detection. For the ordinary evaluation techniques for food, such as artificial sensory evaluation, colorimetric methods [3,4], high performance liquid chromatography-mass spectrometry (HPLC-MS) [5,6,7], and so on, the artificial sensory evaluation of food has the disadvantages of strong subjective factors. Colorimetric method, HPLC-MS, and some other instruments can provide some digital indicators for the quality inspection of sample, but for the evaluation of food, the direct feeling of human beings is taste information, the electronic tongue can detect the comprehensive taste information of the sample in terms of freshness, saltiness, sourness, sputum, and bitterness [8,9,10,11].

At present, many scholars have used electronic tongue system to collect and analyze the taste information of substances. For example, classification of different beer samples [12], identification of wine and tea quality [13,14], geographical origin tracing of black fruit wolfberry [15], freshness detection of milk [16], adulteration detection of pork/chicken in mutton, and so on [17]. With the in-depth development of the electronic tongue technology, the detection of materials has become diversified. In some cases, the detection accuracy of electronic tongue does not meet the requirements. Therefore, some scholars have devoted themselves to improving the accuracy of the electronic tongue system [18,19]. Experts used the self-organizing mapping (SOM) statistical method to classify wine samples through the electronic tongue system, which provided better resolution of sample generation than PCA [20]. Wavelet energy characteristics (WEF) were extracted from response signals of electronic tongue, which improved classification accuracy of different grades of Indian black tea (the improvement was 99.75%) [21]. The local discriminant preservation projection (LDPP) model was proposed from the perspective of algorithm. Kernelized extreme learning machine (KELM) classifier model based on LDPP achieved optimal taste recognition performance with an accuracy of 98% [22]. Some researchers have improved the accuracy of fruit juice recognition through the fusion of electronic tongue and electronic nose [23]. To make the electronic tongue system more similar to human perception, some scholars have made relevant researches on intelligent bionic instruments and fuzzy evaluation. They analyzed olive oil fusion data of electronic tongue signal and artificial sensory attribute information through relevant algorithms, and reached a classification accuracy rate of 100%, indicating that the combination of human sensory evaluation and electronic tongue analysis can successfully identify olive oil products [24]. Based on the taste difference between grafted and non-grafted watermelon fruits, the relationship between sensory evaluation and electronic tongue was studied, suggesting that these two measurement methods can complement each other [25]. Some researchers obtained the descriptive characteristics of commercial seasonings through artificial sensory evaluation, and conducted flavoring taste perception experiments with electronic tongues, which proved that there’s a certain correlation between human senses and electronic tongue sensors [26]. Some researchers have scored rice wine through artificial sensory evaluation. According to the score, the electronic tongue can predict most of the relatively acceptable sensory attributes except bitterness [27]. Until now, most of the existing researches are based on the comparison of artificial sensory evaluation to determine the accuracy of the intelligent bionic instrument and the grades of food samples, so that the intelligent bionic instrument can accurately and digitally express the information of samples [28,29,30,31,32]. These studies are important for achieving high-precision biomimetic prediction of samples.

Nevertheless, in terms of food evaluation, there are still differences between the quantitative digitization results of electronic tongue and human’s fuzzy evaluation language of food. To make the detection of electronic tongue closer to human perception, it is urgent to transform the quantitative output of electronic tongue into a qualitative fuzzy evaluation language that conforms to human perception.

For fuzziness research, cloud model is a cognitive model based on probability theory, which studies the transformation between qualitative and quantitative representation. At present, some researchers have completed a rough grade evaluation of the quality of red wine with the standard cloud obtained by the Golden Section as the reference object [33]. This study has made some contributions to the direction of wine quality grade evaluation. Nonetheless, in the current research, there is no good technical means in the field of fuzzy language evaluation. Based on the concept of cloud model, the cloud model of liquor taste information was established with Chinese liquor as the carrier, and the key problem of transform the quantitative output of electronic tongue into qualitative fuzzy evaluation language in line with human perception was solved. Compared with other liquor evaluation technologies, which are time-consuming, expensive and difficult to popularize, the identification results in the digital form. The model established in this paper can not only accurately and quickly distinguish the flavor of liquor, but also output the human perception evaluation language that conforms to the statistical law. That is, the result of each evaluation is to output a fuzzy language similar to human being on the basis of correctly discriminating the flavor of liquor, so that the evaluation techniques for liquor approaches to the human perception mode. The specific steps to establish a cloud model of liquor taste information is shown in Figure A1. After the model was built, when the unknown liquor was evaluated, the flavor of liquor should be predicted first and entered into the model, and then the fuzzy language evaluation of the liquor taste would be output. When choosing prediction algorithm, many algorithms (backpropagation neural network, Random Forests, Extreme Learning Machine, etc.) could achieve the purpose of judging the flavor of liquor. In this paper, support vector machine (SVM) was selected.

## 2. Acquisition of Liquor Taste Information

### 2.1. Materials

According to Chinese national standards, Chinese liquor can be divided into different flavors. According to the production process and flavor characteristics, Chinese liquor can be categorized into jiang-flavor liquor, feng-flavor liquor, nong-flavor liquor and mild-flavor liquor. Therefore, four kinds of Chinese liquors produced in the same year (2018) with similar alcohol concentration (52–55% by volume) were selected as experimental samples. The specific information of these samples, such as liquor name, flavor style, raw material, alcohol content, and manufacturer is shown in Table 1.

### 2.2. Collection of Liquor Taste Information Based on Electronic Tongue System

The SA402B electronic tongue (e-tongue) analysis system produced by INSENT company was used to collect the taste information of liquor. The instrument is mainly composed of a sensor array, an automatic detection system and a data acquisition system. The sensor array consists of five lipid/polymer membrane taste sensors and two reference electrodes. Among the five taste sensors, AAE, CT0, CA0, AE1, and C00 sensors were used to test umami, salty, sour, astringent, and bitter tastes, respectively. When the samples were analyzed, the lipid/polymer membrane on the sensor responded to the non-volatile taste substances in the samples, causing changes in the potential of the working electrode. The taste information of the sample can then be obtained by measuring the potential difference between the working electrode and the reference electrode. Figure 1 shows the SA402B electronic tongue system.

The test and reference sensors were activated for 24 h prior to sample testing. To avoid damaging the electronic tongue instrument, it is necessary to dilute the liquor to 5% alcohol content with appropriate proportion of distilled water (according to the alcohol content of each scented liquor), and the diluted liquor should be stored in sealed glass containers for further usage. Upon sample detection, 40 mL of the diluted liquor was added into the corresponding small measuring cups. After self-test and diagnosis, the sensor was cleaned in the positive and negative cleaning solution for 90 s to remove any adsorbed material on it. Then, it was washed in the reference solution for 120 s (two reference solutions) until it reached equilibrium, and the reference solution potential would be obtained. The potential of the sample solution was obtained after the sensor was soaked in each sample for 30 s. The basic values of umami, sour, salty, bitter, and astringent could be evaluated by the potential difference of different sensors. The sample was washed for 3 s and then immersed in the reference solution for 30 s to measure the aftertaste value of the diluent. After each measurement was completed, the sensors were cleaned automatically and the next set of measurement will be carried out after the cleaning. Five samples of each liquor were used for four measurements, in which 20 experiments were repeated for each group of liquor. Finally, data for a total of 80 samples were obtained. The experimental temperature was maintained at 23 ± 1 °C and the relative humidity was 30 ± 2% RH.

### 2.3. Acquisition of Liquor Taste Information Based on Artificial Sensory Evaluation

For the electronic tongue to output a language with human emotional characteristics, we need to obtain the frequency and use order of liquor taste description words according to human perception. Therefore, the sensory evaluation experiments of liquors with various flavors were carried out in this paper. The liquor samples were labeled before the experiment and prepared for use under the same conditions. We screened 40 young people aged 22–30 with good taste perception ability through taste sensory ability test, and formed a sensory panel (26 males and 14 females). Through the training of sensory words in liquor, they can fully understand the meaning of sensory evaluation words. During the experiment, the samples were poured into clean and dry wine glasses. After gargling, appraiser drank a small amount of samples (~2 mL), tasted them carefully with taste organs, and selected a taste description word that best fits the taste of the liquor (The specific liquor flavor taste description words are shown in Table 2). The detection interval of different flavor liquor samples was 20 min. The liquor taste description words were selected in reference to the national standard of the People’s Republic of China (GB/T 33405-2016) [34].

## 3. Establishment of Liquor Taste Information Cloud Model

### 3.1. Determination of Liquor Flavor Discrimination Algorithm

In this experiment, the electronic tongue was used to detect liquor and 10-dimensional data of taste information was obtained. According to the performance of the electronic tongue, voltage value at the 30th second of the steady-state value after the pretreatment of five taste detection signals and five aftertaste detection signals was taken as the characteristic value of the electronic tongue signal. SVM was adopted to predict the taste information of liquor. In the SVM model, it is essential to select the appropriate ***g*** (kernel function) and ***c*** (penalty factor) to construct and solve the optimization problem [35]:(1)$$\min\frac{1}{2}\sum_{i=1}^{n}\sum_{j=1}^{n}y_{i}y_{j}\alpha_{i}\alpha_{j}K(x_{i},x_{j})-\sum_{i=1}^{n}\alpha_{i}\;s.t.\;\sum_{i=1}^{n}y_{i}\alpha_{i}=0,\;0\leq \alpha_{i}\leq c,\;i=1,\cdots ,l$$

Radial basis was selected as the kernel function in this paper:(2)$$K(x_{i},x_{j})=\exp{\left(-g\left\|x_{i}-x_{j}\right\|\right)}^{2}$$

Therefore, it is necessary to adjust relevant parameters (penalty parameter ***c***; kernel function ***g***) to obtain a relatively ideal classification accuracy. In general, the idea of cross-validation (CV) can avoid the occurrence of over-learning and under-learning states, resulting in optimal parameters. In the sense of CV, the optimization of parameters by genetic algorithm (GA) can be conducive to find parameter ***c*** and ***g*** faster and more stable. To improve the performance of the classifier, the SVM optimized by GA(SVM(GA)) was used to classify four different types of liquor in this study, in which the maximum number of iterations was 200, the population number was 20, the search range of c was $2^{-10}$ to $2^{10}$, and the search range of g was $2^{-10}$ to $2^{10}$. In this experiment, each group of fragrant liquor was repeated 20 times. In the case where the ratio of training set to test set is 7:3, 14 groups were randomly selected as the training set and the remaining six groups were used as the prediction set in each liquor data. The classification results of the electronic tongue liquor data are shown in Figure 2.

Figure 2a illustrates the parameter optimization fitness curve, and Figure 2b is the accuracy classification result of electronic tongue data. To avoid overfitting and underfitting, the highest accuracy of cross-validation of training set is taken as the fitness function. In the process of parameter optimization in Figure 2a, the blue and red circles represent the average fitness and the best fitness of each iteration, respectively. When the best fitness was the highest, the corresponding optimal parameters ***c*** and ***g*** were output. Therefore, when the highest accuracy of five cross-validation of the training set was 100%, the optimal parameters of ***c*** and ***g*** were separately 24.2335 and 0.00066757. On this basis, SVM prediction model was built to train and predict the liquor data, and the classification prediction results in Figure 2b were obtained. Where the blue circles and red circles represent the classification predicted by SVM and the actual category of the data, respectively. It can be seen that the prediction classification accuracy reached 100%. The results show that the SVM(GA) has a positive impact on the prediction performance of the model. In this method, different types of liquor can be predicted well. Therefore, SVM(GA) was used to discriminate liquor flavors, and the results were substituted into the cloud model to obtain the final fuzzy evaluation.

### 3.2. The Concept of the Cloud Model

It has always been difficult for researchers to convert the quantitative digital output of electronic tongue into the final fuzzy evaluation language. However, the cloud model could be an effective tool for qualitative and quantitative conversion [36], which can reflect the randomness of representative sample points and the uncertainty of their membership degree, providing a basic method for this study. Therefore, we have built cloud drop map of liquor taste information to complete the conversion of liquor taste information to fuzzy cloud drops (cloud drops in the cloud model can be regarded as liquor taste information points). The cloud model includes forward cloud generator and backward cloud generator. Forward cloud generator has universal adaptability, where it can convert qualitative concepts into quantitative values. In contrast, the backward cloud generator is a process of transforming quantitative values into qualitative concepts, and can obtain the digital characteristics (Expected value ($E_{X}$), Entropy ($E_{n}$), and Hyper entropy ($H_{e}$)) that reflect the taste information of liquor.

In this study, $E_{X}$ is the average point coordinates of all cloud droplets, reflecting the central position of liquor taste information. $E_{n}$ reflects the numerical range ambiguity of the liquor flavor. $H_{e}$ represents the uncertainty in the measurement of the conceptual ambiguity of liquor flavor. The specific steps of the reverse cloud generator are as follows:

First, calculate the x average of each set of data samples entered $\bar{x}$, then calculate the absolute center distance D of the first-order sample according to the mean of sample array. Then, calculate the sample variance $S^{2}$ of this group of data. Finally, $E_{X}$ of the liquor sample is equal to the mean value of the liquor taste information, $E_{n}$ and $H_{e}$ of the liquor sample is equal to ambiguity and uncertainty of the liquor taste information respectively, The formula are as follows.
(3)$$E_{X}=\bar{x}$$
(4)$$E_{n}=\sqrt{\frac{\pi }{2}}•D$$
(5)$$H_{e}=\sqrt{S^{2}-E_{n}^{2}}$$

According to the formula of the backward cloud generator, $E_{X}$, $E_{n}$, and $H_{e}$ of m-dimension obtained from liquor data could be used as the input for the m-dimension forward cloud generator, and N one-dimensional normal random numbers with expected value $E_{n_{i}}$ and variance $H_{e_{i}}{}^{2}$ are generated. Repeat this step to make i = 1, 2 …, m. On this basis, it can be regenerated into a one-dimensional normal random number $x_{i}$ with expected value $E_{x_{i}}$ and variance $E_{\mathrm{nn}}{}_{i}\prime^{2}$. Repeat this step m times so that i = 1, 2, …, m. After that, calculate the determinacy of each sample point on the concept of liquor flavor:(6)$$\mu =\exp\sum_{i=1}^{m}\left[-\frac{{\left(x_{i}-E_{x_{i}}\right)}^{2}}{2E_{\mathrm{nn}}{}_{i}\prime^{2}}\right]$$
$\left\{x_{1},x_{2},\cdots x_{m}\right\}$ with determinacy (μ) forms a cloud droplet in the universe.

By repeating the above steps for N times, N m-dimensional cloud droplets can be obtained, and the liquor flavor taste information was finally expressed.

### 3.3. Liquor Taste Information Cloud Drop Point Acquisition

Due to the high dimension of response signal in the electronic tongue, other than the characteristic information of liquor, it also contains some redundant information, which will interfere with data processing. To reduce the processing difficulty of the system (reduced processing time and enhanced data visualization), PCA was used to reduce the dimension of the training set data. The several comprehensive indicators it finds reflect the information of the original variables as much as possible and are not related to each other. The accumulated variance contribution rate of the first two-dimensionality data reached 0.9825 (>0.85) after the data were extracted, PC1 was 0.9424 and PC2 was 0.0401. Thus far, the original information of electronic tongue was successfully expressed. Therefore, the two-dimensional cloud model can be built to express the taste information of liquor. PC1 and PC2 were taken as two-dimensional data and input into the backward cloud generator to obtain digital characteristics reflecting liquor taste information, namely, $E_{X}$, $E_{n}$, and $H_{e}$. The obtained characteristic values of different flavor liquors are shown in Table 3.

As in Table 3, PC1 and PC2 are the first and second main components of the original data, respectively. It can be seen that the $E_{X}$ value of the jiang-flavor liquor was similar to that of the nong-flavor liquor, and the expected value of the feng-flavor liquor was quite different from the expected value of other flavor liquors. As a whole, there was little difference in the $E_{n}$ values of the four kinds of liquors, thus the fuzziness of the four kinds of liquors was similar, and the range of cloud droplets had little difference. The $E_{X}$, $E_{n}$, and $H_{e}$ of four kinds of liquor with different flavor were input into the two-dimensional forward cloud generator to recover the aroma evaluation results respectively. The four groups of restored liquor taste information are shown in Figure 3.

In Figure 3, PC1 and PC2 are the two-dimensional coordinates of the liquor information model, respectively, and the vertical axis is the membership degree of cloud droplet, which represents the certainty of the liquor flavor. The plane consisted of PC1 and PC2, representing the overall taste information of each fragrant liquor. In Figure 3, red represents the jiang-flavor liquor cloud droplets, blue represents the feng-flavor liquor cloud droplets, green represents the nong-flavor liquor cloud droplets, and purple represents the mild-flavor style liquor cloud droplets. The cloud droplets of each liquor were consistent with the characteristics of peak and fat tail. Among the four different flavor types of liquor, only the cloud droplets of jiang-flavor liquor and nong-flavor liquor were partially overlapped, indicating that their taste information had certain similarities. The cloud drop result for feng-flavor liquor was far away from other fragrant liquors, therefore, it can be clearly distinguished from other fragrant liquors in terms of taste.

### 3.4. Frequency of Words in Liquor Taste Information

In the artificial sensory evaluation experiment, the number of times that five words were selected by examiners in each group of experiments was counted, and the corresponding frequency of each word was obtained. Meanwhile, the frequencies were arranged from large to small, denoted as $n_{1},n_{2},n_{3},n_{4},n_{5}$ ($n_{1}\geq n_{2}\geq n_{3}\geq n_{4}\geq n_{5}$). For example, in the experiment with jiang-flavor liquor, “Fully mellow” was chosen by 20 people, “Elegant and delicate” was chosen by 10 people, “Full bodied” was chosen by four people, “Long aftertaste” was chosen by four people, and “Coordination” was chosen by two people. According to the statistical analysis, the frequency of $n_{1}$, $n_{2}$, $n_{3}$, $n_{4}$, and $n_{5}$ was 50%, 25%, 10%, 10%, and 5%, respectively. The statistical results of each group are shown in Figure 4.

As can be seen from Figure 4, in the taste evaluation experiment of jiang-flavor liquor, $n_{1}$ = 50%, $n_{2}$ = 25%, $n_{3}$ = 10%, $n_{4}$ = 10%, and $n_{5}$ = 5%, indicating that half of the appraisers believed that the “Fully mellow” in jiang-flavor liquor were more consistent with their taste sensation, whereas only 5 percent of appraisers thought that “Coordination” was closer to the taste experience. In the experiment with feng-flavor liquor, the difference in the frequency of each word selected was relatively small, with $n_{1}$ being 35%, indicating that more than one-third of the appraisers thought “Sweet and cool” can best express their taste perception. In the nong-flavor liquor experiment, “Soft and sweet” was selected and the ratio was 40% ($n_{1}$), indicating that majority of the appraisers thought the characteristics of nong-flavor liquor was sweet, and in the experiment with mild-flavor liquor, the frequencies of taste perception “Pure fragrance” and “Long aftertaste” were both 30%, indicating that these two words can best express the taste characteristics of mild-flavor liquor.

### 3.5. Correlation between the Range of Liquor Taste Information Cloud Droplets and Evaluation Words

According to the cloud drop group based on the frequency of the words selected, the corresponding elliptical ring area in the top view of the cloud drop area was calculated. Wherein, according to Figure 4, the frequencies of the evaluation word were arranged from large to small, corresponding to the different areas of the cloud drop from the center to the edge. The derivation process of the evaluation words corresponding to the cloud drop area mainly included three parts: the contribution of the cloud droplet group to the qualitative concepts, the correlation of words in the cloud droplet central areas, and the correlation of words in the cloud droplet ring areas.

#### 3.5.1. Contribution of Cloud Droplet Groups to Qualitative Concepts

Clouds are composed of many cloud droplet groups, cloud droplet groups are composed of many cloud droplet points. Each cloud droplet point is a point mapped from qualitative concept to numbered domain space, so they all make contributions to the determination of qualitative concept. Among them, the contribution rate ΔC of the element on any intervals Δx in cloud X to the qualitative concept $\tilde{A}$ is [37]
(7)$$\Delta C\approx \mu \tilde{A}{\left(x\right)}^{*}\frac{\Delta x}{\sqrt{2\pi }En}$$
where En is the sample entropy value. Therefore, the total contribution rate C of all elements representing concept $\tilde{A}$ in the universe is
(8)$$C=\int_{-\infty }^{+\infty }\int_{-\infty }^{+\infty }\frac{1}{2\pi En_{1}En_{2}}e^{-\frac{1}{2}[\frac{{\left(x-Ex_{1}\right)}^{2}}{En_{1}^{2}}+\frac{{\left(y-Ex_{2}\right)}^{2}}{En_{2}^{2}}]}=1$$
where $Ex_{1}$ is the expected value of the first dimensional data in the two-dimensional cloud model, $En_{1}$ is the entropy of the first dimensional data in the two-dimensional cloud model, $Ex_{2}$ is the expected value of the second dimensional data in the two-dimensional cloud model, and $En_{2}$ is the entropy of the second dimensional data in the two-dimensional cloud model.

According to the calculation, the contribution rate of cloud droplets to the qualitative concept in different intervals can be obtained. On the contrary, the corresponding cloud droplet region can be calculated by the required proportion. Therefore, the region of cloud drop group can be divided according to the frequency of words needed in this paper.

#### 3.5.2. Correlation of Words in the Cloud Droplet Central Areas

As we mentioned in Section 3.4, $n_{1}$ represented the highest frequency of a word selected, whereas the word corresponded to the location of the central region of this cloud droplet group, which can then be calculated as follows.

According to the characteristics of the two-dimensional normal distribution cloud model, the frequency of a certain word in the one-dimensional normal distribution is $\sqrt{0.01\times n_{1}}$, if the location range of the word in one-dimension is $\left[Ex-k_{1}En,Ex+k_{1}En\right]$, the probability of *X* falling in the interval $\left(-\infty ,Ex+k_{1}En\right)$ is
(9)$$P(X<Ex+k_{1}En)=\frac{1}{2}+\frac{\sqrt{0.01\times n}}{2}\approx \alpha$$
where $Ex+k_{1}En$ is the right boundary value of the probability that the word has a probability of $\sqrt{0.01\times n}$ in the one-dimensional normal distribution. According to the standard normal distribution table, the probability of α corresponds to the coordinate value of a in the standard normal distribution. The standardized formula is
(10)$$\delta ∼N(\mu ,\sigma^{2})\to \eta =\frac{\delta -\mu }{\sigma }∼N(0,1)$$
where μ is the expected value, σ is the variance. According to Formula (10), the general normal distribution was normalized:(11)$$a=\eta =\frac{Ex+k_{1}En-Ex}{En}=k_{1}$$
where Ex is the expected value and $k_{1}$ is equal to a in the standard ortho-distribution table. Therefore, the range of cloud drop group corresponding to this certain word is
(12)$$\frac{{\left(x-Ex_{1}\right)}^{2}}{{\left(aEn_{1}\right)}^{2}}+\frac{{\left(y-Ex_{2}\right)}^{2}}{{\left(aEn_{2}\right)}^{2}}\leq 1$$

#### 3.5.3. Correlation of Words in the Cloud Droplet Ring Areas

When the proportion of a certain word in a certain flavor liquor is $n_{i}$ (i=2,3,⋯,5), the probability that the sum of the previous words occupying all the words is $\sum_{k=1}^{k=i-1}n_{k}$, and the sum of the cloud droplets covered by previous words is ${\left(x-Ex_{1}\right)}^{2}/{\left(bEn_{1}\right)}^{2}+{\left(y-Ex_{2}\right)}^{2}/{\left(bEn_{2}\right)}^{2}\leq 1$, similarly, the cloud drop area occupied by this word can be calculated. Providing that the word is located in the one-dimensional regions of $\left[Ex-k_{i}En,Ex-bEn\right]$ and $\left[Ex+bEn,Ex+k_{i}En\right]$, the probability that X falls into the interval $\left(-\infty ,Ex+k_{i}En\right)$ is
(13)$$P(X<Ex+k_{i}En)=\frac{1}{2}+\frac{\sqrt{0.01\times \sum_{k=1}^{k=i}n_{k}}}{2}\approx \beta$$

According to the standard normal distribution table, the probability of β corresponds to the coordinate value of c in the standard normal distribution, the standardized formula is
(14)$$\delta ∼N(\mu ,\sigma^{2})\to \eta =\frac{\delta -\mu }{\sigma }∼N(0,1)$$
where μ is the expected value and σ is the variance. According to the Formula (14), general normal distribution is normalized:(15)$$c=\eta =\frac{Ex+k_{i}En-Ex}{En}=k_{i}$$
where $Ex+k_{i}En$ is the right boundary value of the probability of the word in the one-dimensional normal distribution and $k_{i}$ is equal to c in the standard ortho-distribution table. Therefore, the range of cloud drop group corresponding to this word is
(16)$$\frac{{\left(x-Ex_{1}\right)}^{2}}{{\left(cEn_{1}\right)}^{2}}+\frac{{\left(y-Ex_{2}\right)}^{2}}{{\left(cEn_{2}\right)}^{2}}\leq 1\;\mathrm{and}\;\frac{{\left(x-Ex_{1}\right)}^{2}}{{\left(bEn_{1}\right)}^{2}}+\frac{{\left(y-Ex_{2}\right)}^{2}}{{\left(bEn_{2}\right)}^{2}}>1$$

#### 3.5.4. Correlation Result of Cloud Droplet Areas and Evaluation Words

According to the proportion of the taste sensory words selected by the appraisers in each group of liquors (Figure 4), the taste information cloud droplets of liquor were divided into corresponding areas, and the results are shown in Figure 5.

In Figure 5, different colors represented different flavor types of liquor. In the test, SVM was used to judge the liquor flavor category, and then the location of cloud droplets was determined by plugging the category into cloud generator. Therefore, the repletion areas of the two liquor flavor in Figure 5 did not affect the correlation between coverage areas and evaluation language. The specific relationship between the cloud model areas and the liquor evaluation words is shown in Table A1. To avoid repeated words in the final output, MATLAB was used to make the words corresponding to the output of the area appear once when multiple cloud drops fall into the same area. The order of output words in evaluation language was arranged from center to periphery (the proportion of words from high to low) according to the position of ellipse region.

## 4. Results of Fuzzy Evaluation of Liquor Flavor

After the establishment of the model, the as mentioned four types of liquor were tested. Two groups of data were randomly selected for each flavor type of liquor obtained by electronic tongue, and eight groups of liquor experimental data were evaluated. The specific evaluation process was shown in Figure A2. When a group of liquor data was input, the flavor type of liquor was firstly determined by SVM, and then the characteristic values $E_{X}$, $E_{n}$ and $H_{e}$ were determined by the flavor type of liquor. These characteristic values were substituted into two-dimensional forward cloud generator to control the output of five cloud drop points. The fuzzy evaluation of liquor flavor can be obtained by connecting the words in the corresponding cloud droplet area. Figure 6 shows the situation when the cloud droplets of eight groups fall into the corresponding areas.

Figure 6 displayed the results of cloud droplets dripping into the region of four different flavor types of liquors, in which (a,b) correspond to jiang-flavor liquor, (c,d) correspond to feng-flavor liquor, (e,f) correspond to nong-flavor liquor, and (g,h) correspond to mild-flavor liquor. It can be seen from Figure 6a,b that both are taste cloud drops of jiang-flavor liquor. The cloud droplets of the two experiments fell into the jiang-flavor cloud droplets area, indicating that the predicted flavor of the liquor is correct. Three cloud drop points fell into the central ring area in both experiments. The shows that the higher probability of vocabulary in the center area. And in two experiments, the cloud drop points occupied regions 1, 2, and 5 in Figure 6a and the cloud point occupied regions 1, 3, and 4 in Figure 6b, which also shows that the final output language of the two experiments is different. The results were the same for other flavor liquors. Five cloud droplets of each flavor liquor fell within the ranges of corresponding cloud model, which reflects the correctness of the predicted flavor type. The different positions of five cloud droplets in each experiment indicate that the output is fuzzy. Eight groups of liquor experimental data are shown in Table 4.

Table 4 lists the information of actual flavor, predicted flavor, cloud droplets areas and evaluation language. It can be seen that the predicted flavor of four different flavor types of liquor was consistent with the actual flavor, displaying the good prediction ability of the established model. The different evaluation languages of liquor with the same flavor type reflected the fuzziness of the evaluation results.

In comparison with the results of human sensory evaluation, taking jiang-flavor liquor as an example, the words selected in human sensory evaluation results according to their frequencies were “Fully mellow”, “Elegant and delicate”, “Full bodied”, “Long aftertaste”, and “Coordination”. Among them, the proportion for “Fully mellow” was 50%, whereas the frequency of other words were not much different. In Table 3, the word “Fully mellow” appeared the most in the output results of both experiments a and b, whereas other words appeared only once in a or b, which was consistent with the proportion of words selected by artificial sensory evaluation in Figure 4. Similarly, for feng-flavor liquor, there was no significant difference in the proportion of selected words in the results of human sensory evaluation, among which the proportion for “Sweet and cool” was the highest (35%), and the proportion of “Mellow fullness” was the lowest (5%). In Table 3, the word “Sweet and cool” appeared in both experiments c and d, indicating that the word “Sweet and cool” had a higher probability than other words. However, the absence of the word “mellow and fullness” in both experiments suggests that “mellow and fullness” was unlikely to represent the flavor of liquor, and the results were still consistent with the results of the artificial sensory evaluation. As for nong-flavor liquor and mild-flavor liquor, the results were also in line with the results of artificial sensory evaluation. Therefore, it is proved that the results of this model are similar to those of human perception, and the model is suitable for the evaluation of liquor flavor.

## 5. Conclusions

The electronic tongue system can distinguish foods effectively. However, there are still differences between the fuzzy language of human sensory evaluation and electronic tongue system results. The expansion of the cloud droplet map to language were completed by setting up cloud model for the taste information of liquor in this paper, so that the electronic tongue can output a fuzzy language that is closer to the human perception habits, and achieve qualitative fuzzy evaluation of different flavor liquor. First, SVM was used to identify different flavor types of liquor based on the taste information of liquor collected by the electronic tongue (classification accuracy was 100%), indicating that SVM could be used to identify liquor flavor types in this study. Second, based on the cloud model generator and artificial sensory evaluation experiment, the cloud model of liquor taste information was built successfully, thus completing the correlation between liquor evaluation words and cloud droplet group. Finally, fuzzy semantic evaluation was successfully obtained for all four kinds of liquor, and the test results of each liquor were in line with the experimental results of artificial sensory evaluation and could accurately discriminate the samples, which proved that the method developed in this study was successful.

Based on the electronic tongue system, an output method that conforms to the human perception model evaluation was developed. It can discriminate unknown flavor liquor accurately and output evaluation language in line with human perception, making the original mechanical evaluation closer to human perception. This method is conducive to simulate sample evaluation of human perception systems.

## Figures and Tables

**Figure 1 sensors-20-00686-f001:**
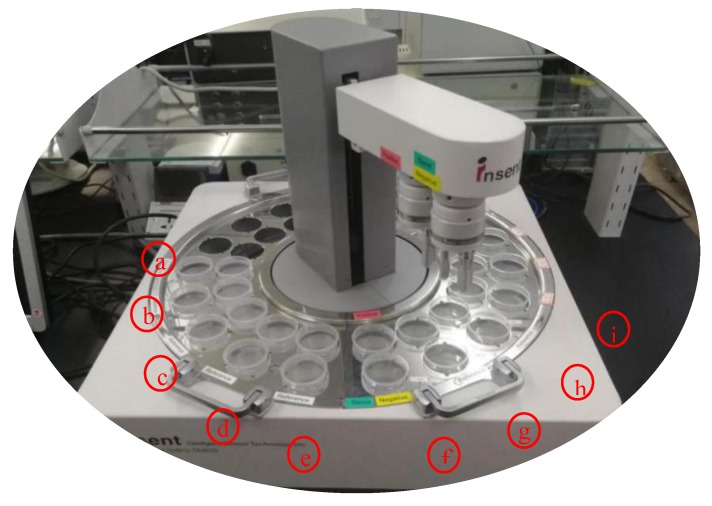
The SA-402B e-tongue system (a is used to measure the aftertaste value, b–c is used to quickly clean the sample, d–e is used to clean the positive and negative solution, f is the positive and negative cleaning solution, g is applied for sensor calibration, h is used for sensor reset, and i is liquor sample).

**Figure 2 sensors-20-00686-f002:**
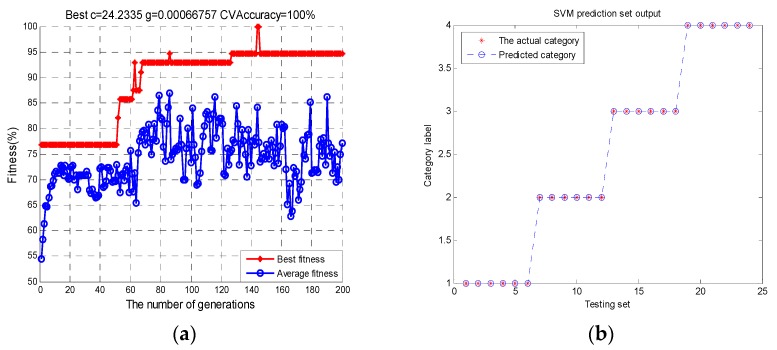
Classification process for liquor taste information with GA-SVM: (**a**) The parameter optimization fitness curve. (**b**) The classification result.

**Figure 3 sensors-20-00686-f003:**
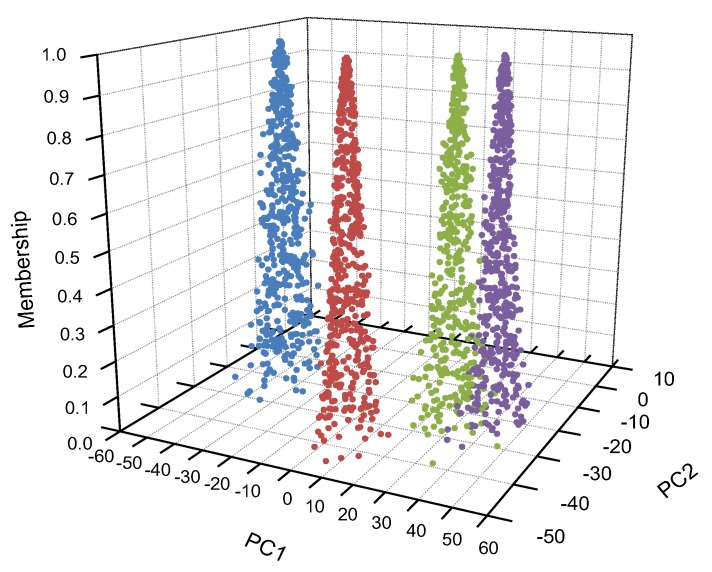
Cloud drop results of four different flavor types of liquor (●: jiang-flavor style, ●: feng-flavor style, ●: nong-flavor style, ●: mild-flavor style).

**Figure 4 sensors-20-00686-f004:**
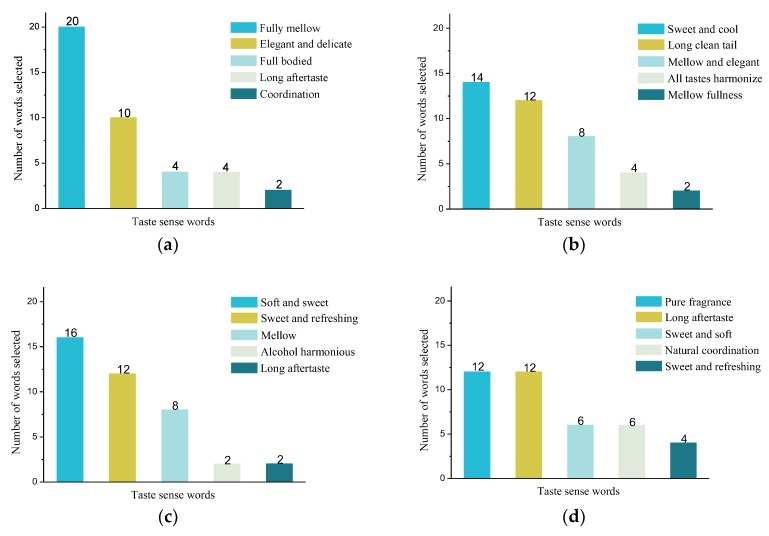
The proportion of liquor taste words selected by examiners: (**a**) jiang-flavor style, (**b**) feng-flavor style, (**c**) nong-flavor style, and (**d**) mild-flavor style.

**Figure 5 sensors-20-00686-f005:**
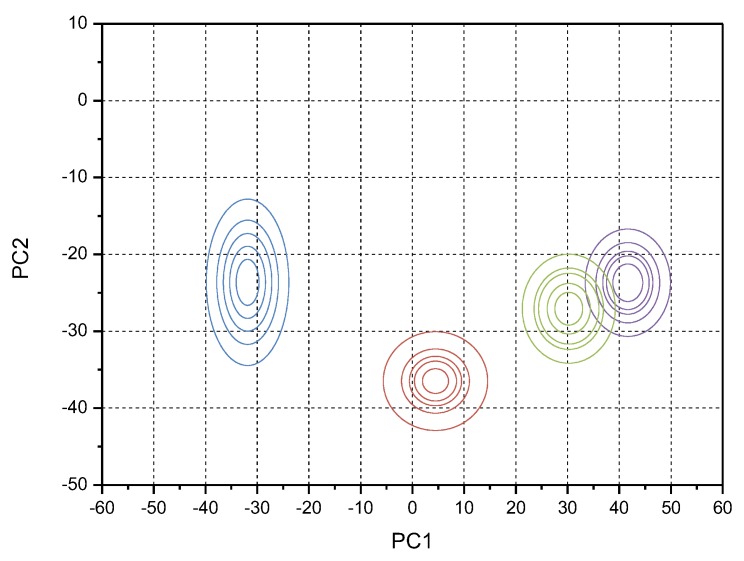
Liquor taste information cloud model division result (purple: jiang-flavor style; blue: feng-flavor style; green: nong-flavor style; brown: mild-flavor style).

**Figure 6 sensors-20-00686-f006:**
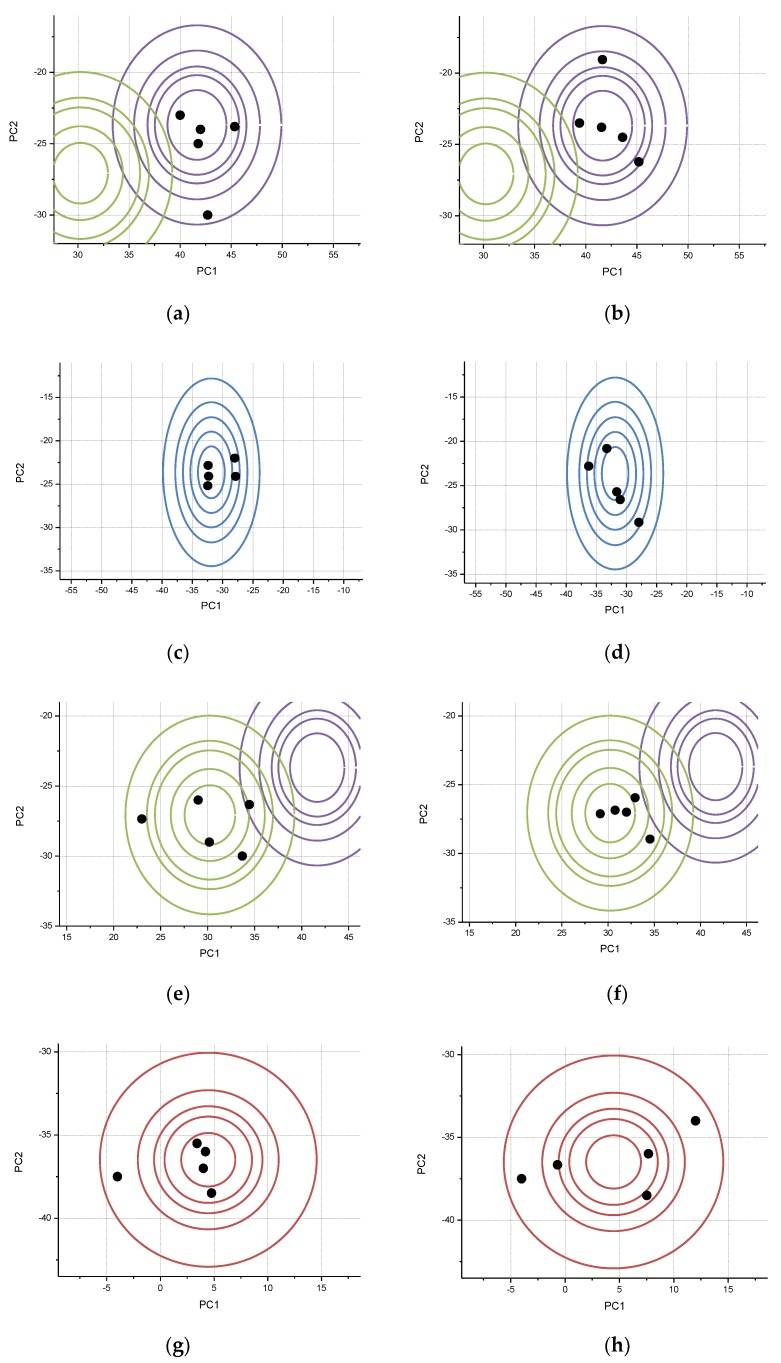
Cloud drop test of 8 groups of liquor data: (**a**) jiang-flavor style-1; (**b**) jiang-flavor style-2; (**c**) feng-flavor style-1; (**d**) feng-flavor style-2; (**e**) nong-flavor style-1; (**f**) nong-flavor style-2; (**g**) mild-flavor style-1; (**h**) mild-flavor style-2.

**Table 1 sensors-20-00686-t001:** Characteristics of sampled liquor.

Liquor Name	Flavor	Raw Material	Alcohol Content (% vol)	Manufacturer
Sauce incense private 1979	jiang	Water, sorghum, wheat	53	Shijia Wine Industry Co., Ltd.
Xifeng wine	feng	Water, sorghum, barley, wheat, peas	55	Shanxi Xifeng Wine Co., Ltd.
Sealed puree wine V60	nong	Water, sorghum, wheat, rice, corn, glutinous rice	52	Ziyunting Wine Co., Ltd.
Red Star Erguotou	mild	Sorghum, water, corn, barley, peas	52	Beijing Red Star Co., Ltd.

**Table 2 sensors-20-00686-t002:** Liquor taste description vocabulary.

Flavor	Liquor Taste Description Words
jiang	Elegant and delicate, Fully mellow, Full bodied, Long aftertaste, Coordination
feng	Mellow fullness, Sweet and cool, Mellow and elegant, All tastes harmonize, Long clean tail
nong	Alcohol harmonious, Sweet and refreshing, Soft and sweet, Long aftertaste, Mellow
mild	Pure fragrance, Sweet and soft, Natural coordination, Sweet and refreshing, Long aftertaste

**Table 3 sensors-20-00686-t003:** Characteristic values of different flavor liquors.

Flavor	Ex		En		He	
	PC1	PC2	PC1	PC2	PC1	PC2
jiang	41.6501	−23.6840	2.7506	2.3288	1.2743	1.1007
feng	−31.8733	−23.6339	2.6604	3.6078	0.3852	1.8173
nong	30.2356	−27.0657	2.9954	2.3639	1.3505	1.6552
mild	4.4596	−36.4820	3.3641	2.1433	0.5800	0.7641

**Table 4 sensors-20-00686-t004:** Results of 8 groups of liquor experimental data.

Serial Number	The Actual Flavor of Liquor	Predicted Flavor of Liquor	Cloud Droplets Areas	Evaluation Language
a	jiang	jiang	$\frac{{\left(x-41.6501\right)}^{2}}{2.8881^{2}}+\frac{{\left(y+23.684\right)}^{2}}{2.4452^{2}}\leq 1$; $\begin{array}{l} \frac{{\left(x-41.6501\right)}^{2}}{4.1259^{2}}+\frac{{\left(y+23.684\right)}^{2}}{3.4932^{2}}\leq 1\\ \&\frac{{\left(x-41.6501\right)}^{2}}{2.8881^{2}}+\frac{{\left(y+23.684\right)}^{2}}{2.4452^{2}}>1 \end{array}$; $\begin{array}{l} \frac{{\left(x-41.6501\right)}^{2}}{8.2518^{2}}+\frac{{\left(y+23.684\right)}^{2}}{6.9864^{2}}\leq 1\\ \&\frac{{\left(x-41.6501\right)}^{2}}{6.1613^{2}}+\frac{{\left(y+23.684\right)}^{2}}{5.2165^{2}}>1 \end{array}$	This liquor is fully mellow, elegant and delicate, coordination
b	jiang	jiang	$\frac{{\left(x-41.6501\right)}^{2}}{2.8881^{2}}+\frac{{\left(y+23.684\right)}^{2}}{2.4452^{2}}\leq 1$; $\begin{array}{l} \frac{{\left(x-41.6501\right)}^{2}}{4.84^{2}}+\frac{{\left(y+23.684\right)}^{2}}{4.0987^{2}}\leq 1\\ \&\frac{{\left(x-41.6501\right)}^{2}}{4.1259^{2}}+\frac{{\left(y+23.684\right)}^{2}}{3.4932^{2}}>1 \end{array}$; $\begin{array}{l} \frac{{\left(x-41.6501\right)}^{2}}{6.1613^{2}}+\frac{{\left(y+23.684\right)}^{2}}{5.2165^{2}}\leq 1\\ \&\frac{{\left(x-41.6501\right)}^{2}}{4.84^{2}}+\frac{{\left(y+23.684\right)}^{2}}{4.0987^{2}}>1 \end{array}$	This liquor is fully mellow, full bodied, long aftertaste
c	feng	feng	$\frac{{\left(x+31.8733\right)}^{2}}{2.2081^{2}}+\frac{{\left(y+23.6339\right)}^{2}}{2.9945^{2}}\leq 1$; $\begin{array}{l} \frac{{\left(x+31.8733\right)}^{2}}{4.6823^{2}}+\frac{{\left(y+23.6339\right)}^{2}}{6.3497^{2}}\leq 1\\ \&\frac{{\left(x+31.8733\right)}^{2}}{3.4585^{2}}+\frac{{\left(y+23.6339\right)}^{2}}{4.6901^{2}}>1 \end{array}$	This liquor is sweet and cool, mellow and elegant
d	feng	feng	$\frac{{\left(x+31.8733\right)}^{2}}{2.2081^{2}}+\frac{{\left(y+23.6339\right)}^{2}}{2.9945^{2}}\leq 1$; $\begin{array}{l} \frac{{\left(x+31.8733\right)}^{2}}{3.4585^{2}}+\frac{{\left(y+23.6339\right)}^{2}}{4.6901^{2}}\leq 1\\ \&\frac{{\left(x+31.8733\right)}^{2}}{2.2081^{2}}+\frac{{\left(y+23.6339\right)}^{2}}{2.9945^{2}}>1 \end{array}$; $\begin{array}{l} \frac{{\left(x+31.8733\right)}^{2}}{4.6823^{2}}+\frac{{\left(y+23.6339\right)}^{2}}{6.3497^{2}}\leq 1\\ \&\frac{{\left(x+31.8733\right)}^{2}}{3.4585^{2}}+\frac{{\left(y+23.6339\right)}^{2}}{4.6901^{2}}>1 \end{array}$ $\begin{array}{l} \frac{{\left(x+31.8733\right)}^{2}}{5.9593^{2}}+\frac{{\left(y+23.6339\right)}^{2}}{8.0815^{2}}\leq 1\\ \&\frac{{\left(x+31.8733\right)}^{2}}{4.6823^{2}}+\frac{{\left(y+23.6339\right)}^{2}}{6.3497^{2}}>1 \end{array}$	This liquor is sweet and cool, long clean tail, mellow and elegant, all tastes harmonize
e	nong	nong	$\frac{{\left(x-30.2356\right)}^{2}}{2.6959^{2}}+\frac{{\left(y+27.0657\right)}^{2}}{2.1275^{2}}\leq 1$; $\begin{array}{l} \frac{{\left(x-30.2356\right)}^{2}}{4.1636^{2}}+\frac{{\left(y+27.0657\right)}^{2}}{3.2858^{2}}\leq 1\\ \&\frac{{\left(x-30.2356\right)}^{2}}{2.6959^{2}}+\frac{{\left(y+27.0657\right)}^{2}}{2.1275^{2}}>1 \end{array}$; $\begin{array}{l} \frac{{\left(x-30.2356\right)}^{2}}{5.8410^{2}}+\frac{{\left(y+27.0657\right)}^{2}}{4.6096^{2}}\leq 1\\ \&\frac{{\left(x-30.2356\right)}^{2}}{4.1636^{2}}+\frac{{\left(y+27.0657\right)}^{2}}{3.2858^{2}}>1 \end{array}$; $\begin{array}{l} \frac{{\left(x-30.2356\right)}^{2}}{8.9862^{2}}+\frac{{\left(y+27.0657\right)}^{2}}{7.0917^{2}}\leq 1\\ \&\frac{{\left(x-30.2356\right)}^{2}}{6.7097^{2}}+\frac{{\left(y+27.0657\right)}^{2}}{5.2951^{2}}>1 \end{array}$	This liquor is soft and sweet, sweet and refreshing, Mellow, long aftertaste
f	nong	nong	$\frac{{\left(x-30.2356\right)}^{2}}{2.6959^{2}}+\frac{{\left(y+27.0657\right)}^{2}}{2.1275^{2}}\leq 1$; $\begin{array}{l} \frac{{\left(x-30.2356\right)}^{2}}{4.1636^{2}}+\frac{{\left(y+27.0657\right)}^{2}}{3.2858^{2}}\leq 1\\ \&\frac{{\left(x-30.2356\right)}^{2}}{2.6959^{2}}+\frac{{\left(y+27.0657\right)}^{2}}{2.1275^{2}}>1 \end{array}$; $\begin{array}{l} \frac{{\left(x-30.2356\right)}^{2}}{5.8410^{2}}+\frac{{\left(y+27.0657\right)}^{2}}{4.6096^{2}}\leq 1\\ \&\frac{{\left(x-30.2356\right)}^{2}}{4.1636^{2}}+\frac{{\left(y+27.0657\right)}^{2}}{3.2858^{2}}>1 \end{array}$	This liquor is soft and sweet, sweet and refreshing, mellow
g	mild	mild	$\frac{{\left(x-4.4596\right)}^{2}}{2.523^{2}}+\frac{{\left(y+36.482\right)}^{2}}{1.6075^{2}}\leq 1$; $\begin{array}{l} \frac{{\left(x-4.4596\right)}^{2}}{4.0706^{2}}+\frac{{\left(y+36.482\right)}^{2}}{2.5934^{2}}\leq 1\\ \&\frac{{\left(x-4.4596\right)}^{2}}{2.523^{2}}+\frac{{\left(y+36.482\right)}^{2}}{1.6075^{2}}>1 \end{array}$; $\begin{array}{l} \frac{{\left(x-4.4596\right)}^{2}}{10.0923^{2}}+\frac{{\left(y+36.482\right)}^{2}}{6.4299^{2}}\leq 1\\ \&\frac{{\left(x-4.4596\right)}^{2}}{6.56^{2}}+\frac{{\left(y+36.482\right)}^{2}}{4.1794^{2}}>1 \end{array}$	This liquor is pure fragrance, long aftertaste, sweet and refreshing
h	mild	mild	$\begin{array}{l} \frac{{\left(x-4.4596\right)}^{2}}{4.0706^{2}}+\frac{{\left(y+36.482\right)}^{2}}{2.5934^{2}}\leq 1\\ \&\frac{{\left(x-4.4596\right)}^{2}}{2.523^{2}}+\frac{{\left(y+36.482\right)}^{2}}{1.6075^{2}}>1 \end{array}$; $\begin{array}{l} \frac{{\left(x-4.4596\right)}^{2}}{5.0462^{2}}+\frac{{\left(y+36.482\right)}^{2}}{3.2150^{2}}\leq 1\\ \&\frac{{\left(x-4.4596\right)}^{2}}{4.0706^{2}}+\frac{{\left(y+36.482\right)}^{2}}{2.5934^{2}}>1 \end{array}$; $\begin{array}{l} \frac{{\left(x-4.4596\right)}^{2}}{6.56^{2}}+\frac{{\left(y+36.482\right)}^{2}}{4.1794^{2}}\leq 1\\ \&\frac{{\left(x-4.4596\right)}^{2}}{5.0462^{2}}+\frac{{\left(y+36.482\right)}^{2}}{3.2150^{2}}>1 \end{array}$; $\begin{array}{l} \frac{{\left(x-4.4596\right)}^{2}}{10.0923^{2}}+\frac{{\left(y+36.482\right)}^{2}}{6.4299^{2}}\leq 1\\ \&\frac{{\left(x-4.4596\right)}^{2}}{6.56^{2}}+\frac{{\left(y+36.482\right)}^{2}}{4.1794^{2}}>1 \end{array}$	This liquor is long aftertaste, sweet and soft, natural coordination, sweet and refreshing

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
