# Peer review of "Fuzzy Evaluation Output of Taste Information for Liquor Using Electronic Tongue Based on Cloud Model"

_sensors, 2020, doi:10.3390/s20030686_

Round 1

Reviewer 1 Report

Well done.

Author Response

Dear Reviewer,
Thank you for your recognition, your recognition gives us great encouragement for academic writing and scientific effort. Best wish for you.

Reviewer 2 Report

The paper describes an interesting fusing of an electronic tongue and data obtained from artificial sensory evaluation. The paper doesn't have an easy read and more tangible information about the models could be shown. Some corrections should be made:

 - There are some red highlighted in the text

 - Page 3, lines 107 – 118 should not be in the introduction section

 - Figure 2 could be better explained

 - What is the original data of the PCA, i.e. what kind of signal the electronic tongue provide?

 - The authors sometimes write Fig. and sometimes Figure, please provide just one form of the word.

 - Please explain better the cloud drop shown in Figure 6.

 - Figure 6)e there is one data that falls into the overlapped region.

Author Response

Dear Reviewer,

Thank you for the comments concerning our manuscript. According to your constructive comments and suggestions, we have made the following changes to the article.

Point 1:

There are some red highlighted in the text

Answer: Thank you for your advice. Since we are resubmitting, the red part of the previous manuscript is to reply to the reviewer's comments. In the manuscript submitted this time, we cancelled the original red part.

Point 2:

Page 3, lines 107 – 118 should not be in the introduction section

Answer: Thank you for your suggestion. We have deleted and changed it, lines 107 in the text are highlighted in red. 

Point 3:

Figure 2 could be better explained

Answer: Thanks for your suggestion. In the process of parameter optimization in Figure 2(a), it can be seen that the blue circle represents the average fitness of each iteration, the red circle is the best fitness of each iteration, and when the best fitness is the highest, the corresponding optimal parameter c and g will be output. In order to avoid overfitting and underfitting, the highest cross validation accuracy of training set is taken as the fitness function. It can be seen that when the highest cross validation accuracy of training set is 100%, the best parameters of c and g are 24.2335 and 0.00066757 respectively. On this basis, we train and predict the liquor data, and we can get the classification prediction results of Figure 2(b). Among them, the blue circle is the category results predicted by SVM, and the red dot represents the real category of the data. It can be seen that the accuracy of prediction classification reached 100%. We have changed lines 201-205, 208-212 and red highlighted in the text.

Point 4:

What is the original data of the PCA, i.e. what kind of signal the electronic tongue provide?

Answer: Thank you for your advice. We choose SA-402B electronic tongue. When the samples were analyzed, the lipid/polymer film on the sensor in the electronic tongue responds to the non-volatile odor in the sample, and the working electrode potential changes. By measuring the potential difference between the working electrode and the reference electrode, we can get the taste information of the sample to be tested. This taste information is recognized set as the basic flavor. After the sensor is cleaned gently, the potential difference between the working electrode and the reference electrode is measured again as aftertaste. Therefore, the original data of PCA is the potential difference between the working electrode and the reference electrode.

Point 5:

The authors sometimes write Fig. and sometimes Figure, please provide just one form of the word.

Answer: Thank you for your comment. We have unified into ‘Figure’, and marked them in red.

Point 6:

Please explain better the cloud drop shown in Figure 6.

Answer: Thank you very much for your questions. According to Figure 6, we can see the distribution of cloud drops of four flavor liquors through the positive cloud generator. From the areas where these cloud drops are scattered, we can see that each liquor cloud drop is located in the area of its own flavor cloud model, which shows that their scattered flavor areas are correct. From the analysis of the same flavor liquor, there are more cloud drops in the central area. It shows that the probability of words in the central area is higher than that in the peripheral area. And every time the same kind of words are scattered in different areas, it also shows that the cloud drops scattered has a certain fuzziness. Through the discussion and analysis of jiang-flavor liquor in the two experiments, the explanation of Figure 6 is deepened, which is shown in red on line 419-426 in the text.

Point 7:

Figure 6)e there is one data that falls into the overlapped region.

Answer: Thanks for your kind reminder. Because jiang-flavor liquor and nong-flavor liquor have certain similarity in taste, some of their cloud drop areas are overlapped. In order to avoid the confusion of the corresponding words after the cloud drops scattered in the experiment, first we use SVM to judge the flavor information of the input liquor, and then substitute it into the corresponding cloud generator of the flavor type to determine the position of the liquor. Therefore, although the cloud drop point in Figure 6(e) falls in the repeated area, before the cloud drips, it has been judged that its fragrance type is nong-flavor, so the output words are still nong-flavor type words, which is not affected.

Reviewer 3 Report

The manuscript sensors-702120 is very interesting and proposes a more direct and user-friendly use of E-tongue systems, although a simple mathematical data treatment would allow a better understanding of the topic. The work can be accepted after some minor corrections:

1) Sensory panel

A better description of the sensory panel is needed.

Are the panelists trained? Is it an official sensory panel?

How accurate are the outputs of the panel?

What is the panel composition? How many men and how many women and with which ages?

Please clarify.

2) Page 4, line 140 please change "When the samples were detected, the ..." by "When the samples were analyzed, the ..."

3) Figure 1 captions: the abbreviation e-tongue was not introduced.

4) Page 7: Equations (3) to (5) are very well-known and can be removed.

5) For the SVM all the sensors of the E-tongue were used (needed) or not?

Author Response

Dear Reviewer,

Thank you for the comments concerning our manuscript. According to your constructive comments and suggestions, we have made the following changes to the article.

Point 1:

Sensory panel

A better description of the sensory panel is needed.

Are the panelists trained? Is it an official sensory panel?

How accurate are the outputs of the panel?

What is the panel composition? How many men and how many women and with which ages?

Please clarify.

Answer: Thanks for your comment. First, we gathered a lot of young people aged 22-30 to screen their taste perception ability. Three kinds of water solutions with different concentrations were prepared from four basic tastes, namely, acid, sweet, salty and fresh. Then we added a cup of pure water solution, a total of 13 cups of solution, and tested all the evaluators. If the answer is correct more than 10 cups, it means that the sensory ability is in the top 30% of ordinary people. Through screening, we finally selected 40 people who answered more than 10 cups correctly to form a sensory panel (26 men and 14 women). By the train, they fully understand the meaning of liquor sensory evaluation words, so as to carry out artificial sensory evaluation experiments. We've been able to describe the sensory panel better in the manuscript. And lines 161-164 were marked in red in the text.

Point 2:

Page 4, line 140 please change "When the samples were detected, the ..." by "When the samples were analyzed, the ..."

Answer: Thank you for your comment. We have changed it on line 129 and marked it in red.

Point 3:

Figure 1 captions: the abbreviation e-tongue was not introduced.

Answer: Thank you for your kind reminder. We've already introduced the abbreviation e-tongue on line 124.

Point 4:

Page 7: Equations (3) to (5) are very well-known and can be removed.

Answer: Thank you for your comment. We have removed these formulas and highlighted them in red on lines 232-234.

Point 5:

For the SVM all the sensors of the E-tongue were used (needed) or not?

Answer: Thank you for your question. There are five sensors in the electronic tongue sensor array used in this paper (fresh sensor, salty sensor, sour sensor, bitter sensor and astringent sensor). Each sensor could get the basic taste and aftertaste, and the output data was 10 dimensional. In the process of data processing, these data were all applied to the input of SVM, so SVM used all the sensors of electronic tongue in this paper.

This manuscript is a resubmission of an earlier submission. The following is a list of the peer review reports and author responses from that submission.

Round 1

Reviewer 1 Report

The authors have realized revision appropriately. I have just noticed minor points (below), which might be easily corrected:

Table 1, if production year was always 2018, this information does not need to be repeated 4-times, but might be included in the table heading.

New lines 128-144, not "90s", but "90 s" - make always space between value and its unit.

The manuscript is now acceptable for publication.

Reviewer 2 Report

(1) This paper lacks enough novelty. They are using known techniques and instruments solving an old topic.

(2) The literature survey is not complete. Some important literature for the topic have not been cited.

Reviewer 3 Report

I compared the original manuscript with the revised manuscript carefully, but I didn't see any changes to my comments, even the contents in the same page and line numbers of these two manuscripts are the same. In addition, the authors delete my comments to the manuscript (comment 1). My original comments are as follows.

The authors adopt a new method to evaluate the quality of Chinese liquor. Below I listed some detailed comments showing that the article should have more attention during the writing.

The title didn't cover the main information about the manuscript, such as electronic tongue. What are the disadvantages of ordinary evaluation techniques for liquor? What are the advantages of your model? Give more presentation in Introduction. The authors adopt support vector machine (SVM) and genetic algorithm (GA) to predict the type of liquor, but these methods didn't preset in Introduction. In addition, why don't you choose other prediction methods? such as BP neural network, since any nonlinear function can be achieved by a three-layer BP neural network. The authors should have a clearly description about these issues in the Introduction. The cloud model has been applied to the quality evaluation of Chinese red wine, what's your contribution to the quality evaluation of Chinese liquor using cloud model? I don't see the text about Discussion. What is the relationship between your research findings and other scholars? Do you support or oppose their opinions? What's your contribution to the quality evaluation of liquor? The original data should be uploaded as electronic supplementary material.
